# Calcium Alginate Production through Forward Osmosis with Reverse Solute Diffusion and Mechanism Analysis

**DOI:** 10.3390/membranes13020207

**Published:** 2023-02-08

**Authors:** Da-Qi Cao, Kai Tang, Wen-Yu Zhang, Cheng Chang, Jia-Lin Han, Feng Tian, Xiao-Di Hao

**Affiliations:** 1Sino-Dutch R&D Centre for Future Wastewater Treatment Technologies/Key Laboratory of Urban Stormwater System and Water Environment, Beijing University of Civil Engineering and Architecture, Beijing 100044, China; 2Institute of Soil Environment and Pollution Remediation, Beijing Municipal Research Institute of Environmental Protection, Beijing 100037, China; 3Institute of Chemical Engineering, Chemical and Process Engineering, School of Engineering and Physical Sciences, Heriot-Watt University, Edinburgh EH14 4AS, UK

**Keywords:** forward osmosis, calcium alginate, recovery, reverse solute diffusion, thin film composite forward osmosis membrane, electrospinning nanofiber membrane

## Abstract

Calcium alginate (Ca-Alg) is a novel target product for recovering alginate from aerobic granular sludge. A novel Ca-Alg production method was proposed herein where Ca-Alg was formed in a sodium alginate (SA) feed solution (FS) and concentrated via forward osmosis (FO) with Ca^2+^ reverse osmosis using a draw solution of CaCl_2_. An abnormal reverse solute diffusion was observed, with the average reverse solute flux (RSF) decreasing with increasing CaCl_2_ concentrations, while the average RSF increased with increasing alginate concentrations. The RSF of Ca^2+^ in FS decreased continuously as the FO progressed, using 1.0 g/L SA as the FS, while it increased initially and later decreased using 2.0 and 3.0 g/L SA as the FS. These results were attributed to the Ca-Alg recovery production (CARP) formed on the FO membrane surface on the feed side, and the percentage of Ca^2+^ in CARP to total Ca^2+^ reverse osmosis reached 36.28%. Scanning electron microscopy and energy dispersive spectroscopy also verified CARP existence and its Ca^2+^ content. The thin film composite FO membrane with a supporting polysulfone electrospinning nanofiber membrane layer showed high water flux and RSF of Ca^2+^, which was proposed as a novel FO membrane for Ca-Alg production via the FO process with Ca^2+^ reverse diffusion. Four mechanisms including molecular sieve role, electrification of colloids, osmotic pressure of ions in CARP, and FO membrane structure were proposed to control the Ca-Alg production. Thus, the results provide further insights into Ca-Alg production via FO along with Ca^2+^ reverse osmosis.

## 1. Introduction

Alginate is a highly valued polysaccharide and is mainly obtained from seaweed; however, large amounts of fresh water and energy are consumed in the production process [1,2,3]. Recently, alginate was found to account for 15–25% of the total sludge dry weight in aerobic granular sludge (AGS) [2,4]. Therefore, applications of specific granulation of AGS [5,6] were proposed to achieve the targeted alginate production instead of conventional industrial methods that extracted alginate from seaweed [7,8,9,10], thereby contributing to a circular economy and society.

Sodium alginate (SA) is the most commercially produced form of alginate [1,3]. Recently, calcium-alginate (Ca-Alg) that can be prepared from SA by metathesis with appropriate calcium salts has been considered as a novel recyclable alginate material because of its high water-absorbing properties, biocompatibility, and its 3D structure that mimics the extracellular matrix [7,11,12]. These Ca-Alg characteristics promote healing of chronic wounds, thus enabling the widespread use of Ca-Alg in wound dressings [13]. Ca-Alg beads, well-known for their special adsorption ability, are used in producing dyes [14], fungicides like difenoconazole, and insecticides such as nitenpyram [15]. As a biodegradable biomaterial or composite adsorbent, Ca-Alg demonstrates excellent removal/recovery performance for toxic metals (Cu^2+^ and Ni^2+^ ions) [16], valuable trace metals, rare earth ions (e.g., Li, Sr, and La) [17], U (VI) ions [18], and ammonia nitrogen and phosphate [19]. Using Ca-Alg in AGS as a coagulant can retain more biomass and improve wastewater treatment efficiency [20]. Additionally, Ca-Alg enables a good biological microecological environment that can encapsulate microscale zero-valent iron [21], diatom biosilica, and Bacillus subtilis [22] for sewage treatment. Ca-Alg hydrogels can also be used to prepare composite filtration membranes with nanofiber support layers that can be used for molecule/ion separation owing to their good anti-pollution performance [23,24]. Furthermore, Ca-Alg capsules can be used as building materials that can gradually release healing agents without being ruptured under cyclic traffic loading, and are an economically viable, environmentally friendly, and promising self-healing mix material for asphalt pavement [25,26,27].

The alginate solution extracted from AGS is close to 100% water content [3,7,8,9,10,28], which necessitates increased energy consumption when direct drying methods such as spray- and freeze-drying are used. Membrane separation is an effective method for alginate concentration without adding any chemical reagent, thus avoiding secondary pollution [3,10]. During membrane separation methods, forward osmosis (FO), a low-carbon and pollution-free method that does not require external power, has been used recently for determining alginate concentrations [8,9,10]. However, some amount of draw solute from the draw solution (DS) side can diffuse reversely into the feed solution (FS) side because both the active and support layers have a two-way transport ability. The reverse solute diffusion (RSD) reduces the osmotic driving pressure, contaminates the FS, and increases the consumption cost of the draw solute [10,29]. Therefore, RSD in the DS is a recurring problem in FO technology that requires urgent attention [30,31]. Our previous study reported the beneficial effects of RSD in the FO concentration of alginate, where RSD was not deemed as a drawback in the DS [9,10]. However, this phenomenon requires further investigation. Ca-Alg, which is a recycled material, as mentioned previously, can be formed on the FO membrane on the feed side through the reverse infiltration of Ca^2+^ in the DS.

Other studies have also reported the benefits of RSD. For example, Mg^2+^ reverse osmosis was used to form high-purity struvite on the membrane surface, achieving >50% water recovery, >93% ammonium nitrogen removal, and >99% phosphate recovery [32]. RSD can positively influence the system performance using NaHCO_3_ solution as a DS (catholyte) in an osmotic microbial fuel cell through self-buffering with reverse-fluxed sodium bicarbonate [33]. Reverse osmosis using ammonia nitrogen could release extracellular polymers in the microalgae on the FS side during FO dewatering of algae using urine as the DS [34].

Furthermore, a novel FO membrane that can control the reverse solute flux (RSF) of Ca^2+^ and enable a high water flux must be fabricated. However, the conventional commercial FO membranes are manufactured to avoid a trade-off between membrane permeability and selectivity [35], in which the FO membranes are designed to minimize RSF. The nanofiber structure resulting from electrostatic spinning has many excellent characteristics, such as high porosity and permeability, large specific surface area per unit mass, and small inter-fiber pores [36]. A porous nanofiber support layer can reduce the inner concentration polarization and increase water flux by 7- to 8-fold. As such, its use has been proposed for thin film composite FO membranes [37,38,39,40]. Typically, electrostatic spinning materials such as polyethylene terephthalate (PET) [38], polyvinylidene fluoride (PVDF) [41], and polysulfone (PSf) [42] are used. Alternatively, the RSF of the draw solute and the water flux can be controlled by adjusting the preparation parameters of interfacial polymerization (IP) such as the ratio of m-phenylenediamine (MPD) and trimesoyl chloride (TMC) in the active polyamide layer [41].

This study aimed to establish a novel Ca-Alg production technique and analyze its mechanisms and influence factors on FO with RSD. The influence of the DS and FS concentration on the reverse solute flux (RSF) and water flux and the content of Ca^2+^ in recycled materials (Ca-Alg recovery production, CARP) formed on the FO membrane on the FS side were investigated using CaCl_2_ as the draw solute and SA as the feed in FO. Scanning electron microscopy (SEM) and energy dispersion spectroscopy (EDS) were used to analyze the CARP. The thin film composite (TFC) FO membranes with electrospinning nanofibers as a porous support layer and IP polyamide as an active layer were fabricated, and their alginate recovery properties, such as specific RSF and water flux, were compared to those of commercial FO membranes. The basic characteristics of Ca-Alg concentration recovery in the FO process using Ca^2+^ reverse osmosis were obtained.

## 2. Materials and Methods

### 2.1. Materials

SA (molecular weight (MW) = 120–190 kDa) was obtained from Sigma-Aldrich (St. Louis, MO, USA). CaCl_2_ (analytical grade) was purchased from the Chinese Medicine Group Chemical Reagent Co., Ltd., Beijing, China. HCl and HNO_3_ were sourced from Beijing Chemical Reagent Research Institute Co., Ltd., Beijing, China. Polyvinylidene fluoride (PVDF, HSV900, MW ≈ 1000 kDa, Arkema Chemical Co., Ltd., Shanghai, China) and polysulfone (PSf) (MW ≈ 22,000, Merck Co., Ltd., Kenneworth, Rahway, NJ, USA) were used as electrospinning polymers, while N,N-Dimethylformamide (DMF, analytical reagent, 99.5%, Shanghai Yien Chemical Technology Co., Ltd., Shanghai, China) and 1-methyl-2-pyrrolidinone (NMP, 99.5%, Merck) were used as solvents during membrane substrate fabrication. Additionally, 1,3,5-benzenetricarbonyl trichloride (TMC, >98%, Macklin Biochemical Technology Co., Ltd., Shanghai, China), n-hexane (C_6_H_6_, >99%, Macklin Biochemical Technology Co., Ltd., Shanghai, China), and m-phenylenediamine (MPD, >99%, Merck Co., Ltd., Kenneworth, USA) were used to prepare the polyamide (PA) layer of membranes. Cellulose triacetate, with an embedded polyester screen support (CTA-ES) FO membrane (Hydration Technology Innovations Corp., Albany, USA), was used. Ultrapure water (resistivity ≥ 18.2 MΩ) was obtained by purifying tap water using an Arium Comfort II ultrapure water system for laboratory use (Sartorius Corp., Göttingen, Germany). Further, 0.22 μm polyethersulfone disposable filter membrane used in the analytical methods was purchased from Jinteng Experimental Equipment Co., Ltd., Tianjin, China.

### 2.2. Experimental Setup and Technique

The FO experimental setup consisted of the following elements: an FO filter unit (CF042A-FO, Sterlitech Corp., Kent, Auburn, WA, USA), with an FO membrane having an effective area of 42.0 cm^2^, two variable speed gear pumps for recirculating the FS and DS (WT3000-1JA, Baoding Lange Pump Corp., Baoding, China), an electronic balance for measuring the increase in the DS weight (BSA3202S, Sartorius Corp., Gottingen, Germany), a magnetic stirrer (FA2004, Shanghai Shunyu Hengping Scientific Instrument Corp., Shanghai, China) to constantly stir the FS, a computer for data recording, and connecting rubber pipes [10,43].

SA was dissolved and stirred in ultrapure deionized (DI) water at 24 °C for 12 h to obtain the SA solution (1.0–3.0 g/L). CaCl_2_ was added to ultrapure DI water and dissolved by stirring for 3 h. This solution was used as the corresponding DS of CaCl_2_ (1.0–3.0 M). The FO experiment was conducted using 500 mL FS and 1500 mL DS, based on sweep model (flow rates, *u* = 2.5 cm/s along opposite directions on both sides of the membrane) [43] or dead-end model (*u* = 3.0 cm/s for the DS side and *u* = 0 cm/s for the FS side) [44]. The cell with a 5 cm height internal cavity was used on the FS side for the dead-end model. The forward osmosis time of 5 h was used for convenience at present and 10 mL of the FS solution was removed every 1 h and used as a sample. The FS volume was calculated according to the increasing DS mass. Notably, the osmotic pressure of the DS can be considered as a constant in the FO process; therefore, it was not tested.

### 2.3. Preparation of TFC FO Membranes

The electrospinning nanofiber membrane (ENM) substrate was fabricated using an electrospinning machine (ET2535, Yongkang Leye Co., Ltd., Beijing, China). First, electrospinning polymer made of PSf or PVDF (20 wt%) was dissolved in a DMF and NMP solution (volume ratio of 7:3) at a stirring speed of 300 rpm and temperature of 60 °C for 10 h in a magnetic stirring constant temperature water bath (Gaode instrument manufacturing Co., Ltd., Changzhou, China). The polymer solution was then transferred into a plastic syringe. To collect the solution on an aluminum-covered drum, we operated a syringe pump (G20, ID 0.60 mm) at a rate of 0.1 mm/min under high voltage (12.5 kV), placed 16 cm from the drum. The temperature and humidity inside the electrospinning chamber were set to 25 °C and 40%, respectively, in which the electrospinning nanofiber support layer was prepared. Afterwards, the support layer was soaked in an anhydrous ethanol solution for 2 h to improve its hydrophilicity and then stored in ultrapure water.

As pretreatment, the substrates were soaked in ethanol for 2 h prior to interfacial polymerization (IP) for fabrication of FO membranes. Then, a 3.4 wt% MPD and ultrapure water solution was poured onto the nanofiber substrate surface for 5 min, removing the excess solution through nitrogen blowing. Then, the MDP saturated substrate was submerged in a 0.075 wt% TMC solution for 2 min to allow the formation of a PA layer. Finally, the obtained membranes were cured in an oven at 45 °C for 10 min and then stored in ultrapure water for later use, denoted as PSf or PVDF ENM-TFC.

### 2.4. Analytical Methods

A sample solution of 4 mL was taken from the feed and mixed with 0.2 mL HCl solution (50%), and the resultant mixed solution was centrifuged at 4000 rpm for 20 min. Subsequently, the supernatant was collected and added to 0.2 mL HNO_3_ solution (1%) to remove Ca^2+^. Later, the mixed solution was filtered through a 0.22 μm pore diameter membrane and the Ca^2+^ concentration in the filtrate was used to determine the Ca^2+^ concentration in the sample solution. The metal ion concentrations in the aqueous solutions were measured using an inductively coupled plasma spectrometer (ICAP 7000 Series, Thermo Fisher Scientific Co., Ltd., Waltham, MA, USA). Ultrapure water was used as a blank sample with a metal ion concentration of 0 mM.

The CARP after the FO progressed was cut into a 1 cm × 1 cm small piece to observe its surface morphology. Some CARP samples were dried for 48 h by vacuum freeze-drying (FD-1A-50, Beijing Boyikang Experimental Instrument Corp., Beijing, China) and then freeze-dried in liquid nitrogen to prepare their cross-sections. All specimens were sputter-coated with gold for 10 min and then observed under a scanning electron microscope (SEM) (G300, ZEISS Corp., Oberkohen, Germany) operating at 200–7000 V under low vacuum conditions. Elemental analysis of the interface was performed by an energy-dispersive X-ray spectrometer (EDS, Oxford Xplore30) attached to the SEM. Five different points of each sample were observed and digital pictures were subsequently taken.

The pore sizes of the ENMs were determined through mercury porosity experiments (AutoPore Iv 9510, Micromeritics Instrument Co., Ltd., Shanghai, China) and the water contact angles of the ENMs were measured using a contact angle meter (JC2000D4M, Zhongchen Digital Technology Equipment Co., Ltd., Shanghai, China). Finally, the mechanical properties of each sample (shaped into strips of 2 cm × 1 cm) were tested using an electronic universal testing machine (non-metallic direction) (Inspekt table blue 5 kN, Hegewald & Peschke, Dresden, Germany) at a distance of 20 cm and a tensile rate of 2 mm/min.

### 2.5. Evaluation of Forward Osmosis Performance

Figure 1 shows the mass transfer diagram in Ca-Alg production via FO using reverse solute diffusion. Herein, the volume of CARP and the concentration change of Ca^2+^ in DS are negligible. The water flux (*J*_w_) passing through the FO membrane per unit membrane area per unit time at time *t* was calculated by the numerical differentiation of the volume versus time data [10,43], and is expressed as follows:(1)$$J_{w}=\frac{dV_{\mathrm{DS}}}{A_{m}dt}$$
where *V*_DS_ is the DS volume, *A*_m_ is the effective area of the FO membrane, and *t* is the FO time.

The total average RSF of Ca^2+^ (*J*_Ca,av_) in Figure 1, considering Ca^2+^ in the CARP passing through the FO membrane per unit membrane area from time *t* = 0 to *t*, is calculated as follows:(2)$$J_{\mathrm{Ca},\;\mathrm{av}}=\frac{W_{\mathrm{Ca}}}{A_{m}t}=\frac{W_{\mathrm{Ca},r}+C_{t}V_{t}}{A_{m}t}$$
where *W*_Ca_ is the total content of reverse osmosis Ca^2+^ including that in the CARP formed on the FO membrane at time *t*; *W*_Ca,r_ is the Ca^2+^ content in CARP formed on the FO membrane at time *t*; and *C*_t_ and *V*_t_ are the concentration of Ca^2+^ in the FS without considering Ca^2+^ in the CARP and the FS volume at time *t*, respectively.

The RSF of Ca^2+^, *J*_Ca_, without considering that in the CARP passing through the FO membrane per unit membrane area per unit time at time *t*, is calculated as follows:(3)$$J_{\mathrm{Ca}}=\frac{C_{t+t_{0}}V_{t+t_{0}}-C_{t}V_{t}}{A_{m}t_{0}}$$
where $C_{t+t_{0}}$ is the concentration of Ca^2+^ in FS at time (*t* + *t*_0_), $V_{t+t_{0}}$ is the FS volume at time (*t* + *t*_0_), and *t*_0_ is the time interval between two consecutive samplings.

## 3. Results and Discussion

### 3.1. Reverse Solute Diffusion of Calcium Ion for Producing Ca-Alg in Forward Osmosis

The concentrated Ca-Alg was formed on the FO membrane on the feed side as a result of Ca^2+^ reverse osmosis in the DS when the CaCl_2_ solution was used as the DS [10]. As this study proposed a novel Ca-Alg production method, in which Ca-Alg was formed on the FO membrane surface by the interaction between SA and reverse osmosis Ca^2+^ in FO process, the RSF is more helpful than the ratio of RSF to water flux. The Ca^2+^ RSF for various concentrations of FS and DS in the FO process was evaluated to comprehensively discuss the reverse solute diffusion. After 5 h of FO experimentation, the Ca^2+^ content in the FS containing Ca^2+^ in the CARP formed on the FO membrane was measured and the average RSF (*J*_Ca,av_) was calculated using Equation (2). As shown in Figure 2a, an abnormal negative correlation of the average RSF was observed with the increasing CaCl_2_ concentration; additionally, the average RSF decreased with the increase in the Ca^2+^ concentration when 1.0 g/L SA solution was used as the FS; in contrast, the average RSF increased with increasing CaCl_2_ concentrations when ultrapure water was used as the FS. Furthermore, the average Ca^2+^ RSF when SA solution was used as the FS was lower than that when ultrapure water was used as the FS, because of the larger resistance of RSD caused by the CARP formed on the FO membrane surface.

Changes in the RSF during the FO process were investigated to further explore the abnormal results obtained for the negative correlation of the average RSF. The RSF of Ca^2+^ without Ca^2+^ in the CARP for 0–5 h as the FO progressed for 1.0 g/L SA solution, used as the FS with various DS concentrations, is shown in Figure 2b. Generally, the RSF value of Ca^2+^ remains constant during the FO process using ultrapure water as the FS; however, as shown in Figure 2b, the RSF value of Ca^2+^ decreased gradually in the FO process when the SA solution was used as the FS. It should be noted that the lines in Figure 2b simply indicate the variation trends of *J*_Ca_ with FO progress for a clear understanding, because of the lack of a specific functional relationship between *J*_Ca_ and *t* in this study. Furthermore, the initial RSF value of Ca^2+^ (obtained by extrapolation at *t* = 0), as shown in Figure 2b, also differed from the average RSF value of Ca^2+^ in the FO that used ultrapure water as the FS (Table 1), although no membrane fouling was observed at the beginning of the FO.

The average Ca^2+^ content of reverse osmosis (*J*_Ca,av_) after 5 h of FO, with 1.0–3.0 g/L SA solution as the FS and 2.0 M CaCl_2_ solution as the DS, is shown in Figure 3a. As shown in Figure 3a, *J*_Ca,av_ increased with an increase in SA concentration. This increase may be because the utilization efficiency of Ca^2+^ increases and the structure of CARP becomes looser with an increase in the SA concentration [3,7], resulting in an increase in the Ca^2+^ RSD. Moreover, as shown in the circular plots of Figure 3a, the ability of the CARP to attract Ca^2+^ (*W*_Ca,r_/*W*_Ca_) increases with the increasing SA concentration, owing to the more negatively charged alginate anions, which can bond to reverse osmotic Ca^2+^ on the FO membrane surface on the feed side. Furthermore, Figure 3b shows the change in the RSF of Ca^2+^ in the FS without considering Ca^2+^ in the CARP, *J*_Ca_ as the FO progressed; additionally, the RSF of Ca^2+^ decreased gradually for 1.0 g/L SA solution as the FS, while it initially increased and then decreased after 3.5 h for 2 and 3 g/L SA solution as the FS. It is speculated that, as the SA concentration increases, the reverse osmotic Ca^2+^ can react with more alginate ions in the FO process; therefore, the Ca-Alg colloid in the CARP accumulated on the FO membrane surface became larger and was easily eliminated by cross flow, thereby reducing the low resistance of Ca^2+^ reverse osmosis and high RSF. However, at high SA concentrations, the accumulation of CARP on the FO membrane was more significant; therefore, the resistance of Ca^2+^ reverse osmosis increased and the RSF decreased as the FO progressed. Notably, those results obtained were based on the concentration range with 1.0–3.0 g/L and, for a much higher SA solution concentration, the specific results should be further investigated in the future.

### 3.2. Calcium Ions in the Ca-Alg Recovery Production

The Ca^2+^ content in the CARP was evaluated. After FO was completed, the Ca^2+^ content in the FS before and after the elution of CARP on the FO membrane surface was measured. Later, the Ca^2+^ content in the CARP, *W*_Ca,r_ was calculated using the difference in both Ca^2+^ concentrations. The Ca^2+^ content measured in the FS after the elution of CARP, *W*_Ca_ is the total Ca^2+^ content from the reverse osmotic solution containing Ca^2+^ in the CARP formed on the FO membrane. As shown in Table 1, *W*_Ca,r_ and *W*_Ca_ decreased; however, the percentage content of Ca^2+^ in the CARP to total osmosis content of Ca^2+^ (*W*_Ca,r_/*W*_Ca_) increased with increasing DS concentrations for 1.0 g/L SA solution as the FS (Figure 2a), which enhanced the resistance of Ca^2+^ RSD by electrostatic interaction and decreased the salt concentration difference. Furthermore, as shown in Figure 3a, the *W*_Ca,r_/*W*_Ca_ up to 36.28% increased significantly with increasing SA concentrations for 2.0 M CaCl_2_ solution as the DS. Therefore, these results indicated that the Ca^2+^ content in the CARP cannot be neglected while evaluating Ca^2+^ RSD in the Ca-Alg production process via FO with Ca^2+^ reverse diffusion. Furthermore, the Ca^2+^ content rate of total reverse osmosis to SA in the FS (*W*_Ca_/*W*_SA_), as shown in Table 1, indicates that, the lower the Ca^2+^ concentration in the DS and the SA concentration in the FS, the higher the content rate of Ca^2+^ in the concentrated recycled Ca-Alg in the FS. Therefore, optimal concentration conditions of the DS and FS were present to obtain a high water flux and availability of calcium ions while producing the target product Ca-Alg via FO with Ca^2+^ reverse diffusion.

The average water flux, *J*_w,av_, in the FO is also shown in Table 1. The average water fluxes, *J*_w,av_, are 6.63, 9.30, and 9.58 L/m^2^/h in the FO of 1.0 g/L SA solution with 1.0, 2.0, and 3.0 M CaCl_2_ DS, respectively, indicating that the water flux increased slightly with the increasing CaCl_2_ concentration (>2.0 M). The trend of the water flux with the increasing CaCl_2_ concentration differed from that of ultrapure water FO, in which the water flux increased dramatically with the increasing CaCl_2_ concentration (1.0–3.0 M) from 10.66 to 19.29 L/m^2^/h. Therefore, the higher the DS concentration, the faster the decrease in the increase in water flux in the FO of SA solution. However, *J*_w,av_ for 1.0 g/L SA solution as the FS (9.30 L/m^2^/h) was higher than those for 2.0 and 3.0 g/L SA (7.45 and 7.56 L/m^2^/h, respectively), indicating that the water flux is not significantly influenced by the SA concentration. The water flux was dominated by the osmotic pressure difference between both sides of the FO membrane at the beginning of the FO. However, Ca-Alg CARP was formed on the FO membrane on the feed side owing to the interaction between SA and Ca^2+^ reverse osmosis as the FO progressed, thereby resulting in a significant transmembrane osmotic pressure difference. Additionally, the higher the DS concentration, the denser the membrane fouling of the gel layer formed on the FO membrane.

The Ca-Alg gel layer of the CARP formed on the FO membrane surface on the feed side was further analyzed using SEM and EDS. SEM morphology and EDS of Ca and Na elements in the CARP surface toward the feed side formed for 1.0 g/L SA as the FS and 1.0 and 3.0 M CaCl_2_ as the DS to reveal the difference between low and high concentration draw solutes are shown in Figure 4. As shown in Figure 4(a1,b1), the CARP for 3 M DS is more loose than that for 1 M DS, along with a large Ca-Alg colloid formed for 3 M DS, suggesting that, the lower the concentration of DS, the stronger the molecular sieve role of the Ca-Alg formed on the FO membrane surface in the FS for resisting Ca^2+^ RSD. Therefore, for high DS concentrations, the electrostatic repulsion of CARP and the transmembrane salt osmotic pressure difference may dominate the Ca^2+^ RSD. In addition, as shown in Figure 4(a2,b2), the higher the DS concentration, the greater the Ca^2+^ content on the CARP surface, similar to the foregoing results, which may form large Ca-Alg colloids.

Furthermore, Figure 5 shows the SEM and EDS results of Ca and Na elements in the cross-sectional profile of CARP formed via FO with 1.0 g/L SA solution as the FS and 1.0 and 3.0 M CaCl_2_ solution as the DS. As shown in Figure 5(a1,b1), the CARP was thicker for 1.0 M CaCl_2_ than for 3.0 M CaCl_2_, indicating that membrane fouling can be easily eliminated by cross flow for a high DS concentration. As shown in Figure 5(a2,b2), the previous results given in Figure 2a and Table 1, which showed that the Ca^2+^ content in the CARP decreases with the increasing DS concentration, were confirmed.

### 3.3. Ca-Alg Production Using ENM-TFC FO Membranes

The production properties of calcium alginate were also investigated for PSf or PVDF ENM-TFC FO membranes and compared to those obtained for commercial CTA-ES FO membranes, as shown in Table 2. To eliminate the effect of FS flow, a dead-end model (*u* = 3.0 cm/s for the DS side and *u* = 0 cm/s for FS side) was used [44]. The PSf ENM-TFC FO membranes were characterized by higher *J*_w,av_ and *J*_ca,av_ values than both the PVDF ENM-TFC FO and commercial CTA-ES FO membranes. *J*_w,av_ and *J*_ca,av_ values of PSf ENM-TFC FO membranes are higher than those of PVDF ENM-TFC FO membranes owing to the diluent concentration polarization of the support layer membrane caused by the small pore size of PVDF ENM (Figure 6). Therefore, considering the highly hydrophilic (low water contact angle) and elastic modulus (Table 3), PSf ENM is suggested to act as a support layer for FO membranes.

When the PSf ENM-TFC FO membrane was used with 1–3 M CaCl_2_ as the DS and the 1 g/L SA as the FS, *J*_w,av_ was proportional to the concentration of the DS, while *J*_ca,av_ for 2 M CaCl_2_ was higher than that for 3 M CaCl_2_. This could be due to the resistance of the CARP Ca-Alg gel layer. In contrast, when the PSf ENM-TFC FO membrane was used with 2 M CaCl_2_ as the DS and 1–3 g/L SA as the FS, there were little changes to both *J*_w,av_ and *J*_ca,av_, indicating that the SA concentration has only a slight effect on water flux and salt reverse diffusion. In conclusion, using TFC FO membranes with ENM as a support layer, the water flux and reverse solute flux can be increased, allowing control of the properties of the Ca-Alg production through the FO process.

### 3.4. Mechanism Analysis

Based on the above results, we have proposed four mechanisms on the effects of the CARP on the reverse diffusion of Ca^2+^ in the DS in Ca-Alg production via FO with alginate solution as the FS, as shown in Figure 7. The reverse osmotic Ca^2+^ in the DS through the FO membrane reacts with alginate in the FS and the CARP (Ca-Alg) is formed on the FO membrane on the feed side (Figure 7a), which influences the mass transfer of reverse osmotic Ca^2+^ from the DS. The molecular sieve role [45] of the CARP on the FO membrane surface in the feed side (Figure 7b) may play a major role in the low concentration of alginate in the FS. There is a dependence of electrostatic interaction [3,7,8] on the characteristics and number of charges in the gel layer of Ca-Alg CARP colloids (Figure 7c) with the formation of larger Ca-Alg CARP colloids on the FO membrane surface on the feed side because of the increase in alginate concentration and RSD of Ca^2+^. Notably, the driving force of the salt concentration difference between the two sides of the FO membrane decreases because of the accumulation of reverse osmotic salt ions such as Ca^2+^ from the DS and the concentration of alginate ion (A^n+^) on the membrane surface on the feed side (Figure 7d) [46,47]. Designing a novel FO membrane such as the ENM-TFC FO membrane is the most important strategy for Ca-Alg recovery production using an alginate solution FO process with Ca^2+^ reverse solute diffusion (Figure 7e) to regulate water and Ca^2+^ flux.

## 4. Conclusions

In this study, a novel Ca-Alg production method was proposed, in which Ca-Alg in the FS was formed and concentrated via FO with Ca^2+^ reverse osmosis using SA solution as the FS and CaCl_2_ solution as the DS. An abnormal RSD was found, with the average RSF decreasing with increasing CaCl_2_ concentrations (1–3 M) for 1.0 g/L SA solution, while the RSD increased with increasing alginate concentrations (1.0–3.0 g/L) for 2.0 M CaCl_2_ solution. These results were attributed to the formation of the CARP on the FO membrane surface on the feed side. The findings suggested that the Ca^2+^ content in the CARP must be considered to evaluate the actual Ca^2+^ RSF because the percentage of total Ca^2+^ reverse osmosis reached 36.28%. Moreover, optimal concentration conditions of the DS and FS are required to prepare the target product Ca-Alg via FO with Ca^2+^ reverse diffusion. SEM and EDS analyses also verified the existence of CARP and its Ca^2+^ content. In addition, the PSf ENM-TFC FO membrane was characterized by high water and reverse salt flux of Ca^2+^. As such, PSf ENM is proposed as a support layer for TFC FO membranes because of its large pore size and its hydrophilic and mechanical properties. Four mechanisms were proposed to explain the effects of Ca-Alg production via the FO process with the reverse osmosis of Ca^2+^ from DS, such as the molecular sieve role, electrification of colloids, osmotic pressure of ions in CARP, and specific FO membrane structure. Future studies will focus on the alginate extracted from AGS to prepare and recycle Ca-Alg using the proposed method, other calcium salts as draw solutes, membrane fouling, as well as the evaluation of its contribution to the circular economy. 

## Figures and Tables

**Figure 1 membranes-13-00207-f001:**
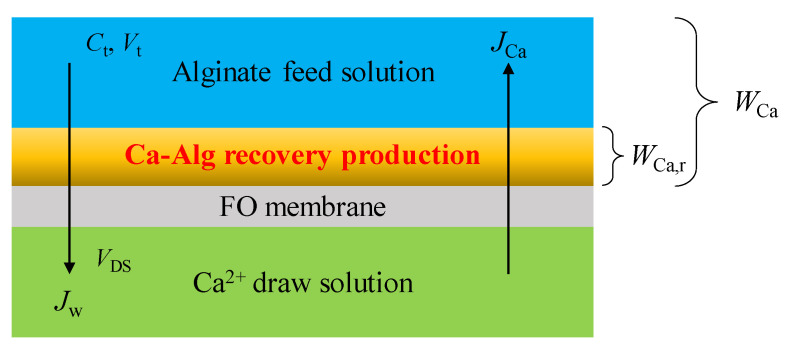
Mass transfer diagram in calcium alginate production via forward osmosis (FO) using reverse solute diffusion. Here, *C*_t_ is the Ca^2+^ concentration in the feed solution (FS) at time *t*; *V*_t_ and *V*_DS_ are the volumes of the FS and the draw solution (DS) at time *t*, respectively; *J*_w_ is the water flux passing through the FO membrane per unit membrane area per unit time at time *t*; *J*_Ca_ is the reverse solute flux (RSF) of Ca^2+^ without considering Ca^2+^ in the CARP passing through the FO membrane per unit membrane area per unit time at time *t*; *W*_Ca_ is the total content of reverse osmosis Ca^2+^ including that in the CARP formed on the FO membrane at time *t*; *W*_Ca,r_ is the Ca^2+^ content in CARP formed on the FO membrane at time *t*.

**Figure 2 membranes-13-00207-f002:**
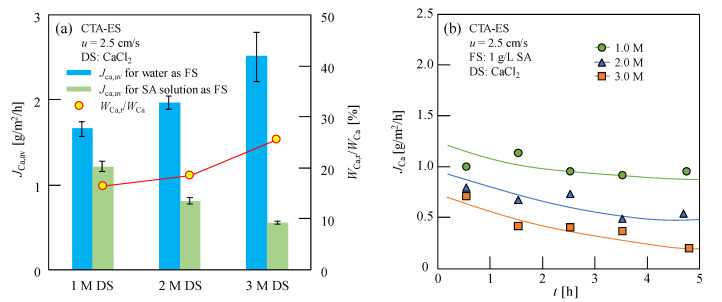
(**a**) Average reverse solute flux (RSF) of Ca^2+^ (*J*_Ca,av_) for ultrapure water and sodium alginate (SA) solution as feed solution (FS) and percentage of Ca^2+^ content in Ca-Alg recovery production (CARP) formed on the forward osmosis (FO) membrane (*W*_Ca,r_) to total Ca^2+^ content (*W*_Ca_) in the FS containing Ca^2+^ in the CARP, *W*_Ca,r_/*W*_Ca_, for SA solution as the FS in FO with various CaCl_2_ solutions as the draw solution (DS); (**b**) RSF of Ca^2+^ without considering Ca^2+^ in the CARP (*J*_Ca_) with the progress of FO. Here, 1.0 g/L SA solution and 1–3 M CaCl_2_ solution were used as the FS and DS, respectively; for the flow rates of 2.5 cm/s, *u* along opposite directions on both sides of the membrane was used; CTA-ES, cellulose triacetate with embedded polyester screen support.

**Figure 3 membranes-13-00207-f003:**
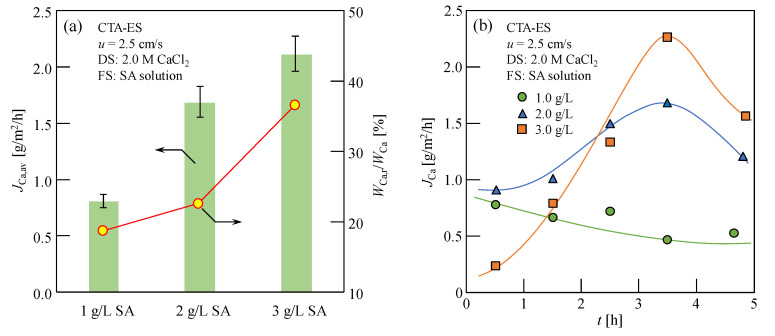
(**a**) Average RSF of Ca^2+^ (*J*_Ca,av_) and percentage of Ca^2+^ content in the CARP formed on the FO membrane (*W*_Ca,r_) to total Ca^2+^ content (*W*_Ca_) in the FS containing Ca^2+^ in the CARP, *W*_Ca,r_/*W*_Ca_ for various SA solutions as the FS; (**b**) RSF of Ca^2+^ without considering Ca^2+^ in the CARP, *J*_Ca_*,* as the FO progressed. Here, 1–3 g/L SA solution and 2.0 M CaCl_2_ solution were used as the FS and DS, respectively; for the flow rates of 2.5 cm/s, *u* along opposite directions on both sides of the membrane was used; CTA-ES, cellulose triacetate with embedded polyester screen support.

**Figure 4 membranes-13-00207-f004:**
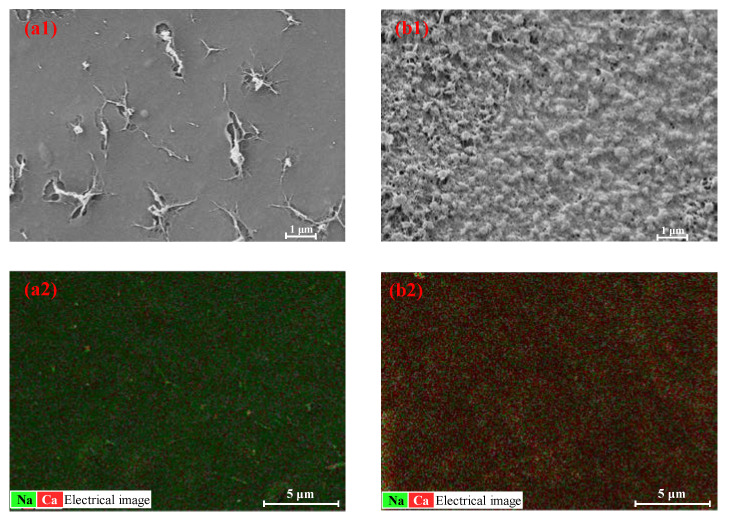
Scanning electron microscopy (SEM) morphology (**a1**,**b1**) and energy-dispersive spectroscopy (EDS) (**a2**,**b2**) of Ca and Na elements in the CARP surface towards the feed side. (**a1**,**a2**) represent the images for CARP formed for 1 M CaCl_2_ as the draw solute, while (**b1**,**b2**) represent the images for CARP formed for 3 M CaCl_2_ as the draw solute.

**Figure 5 membranes-13-00207-f005:**
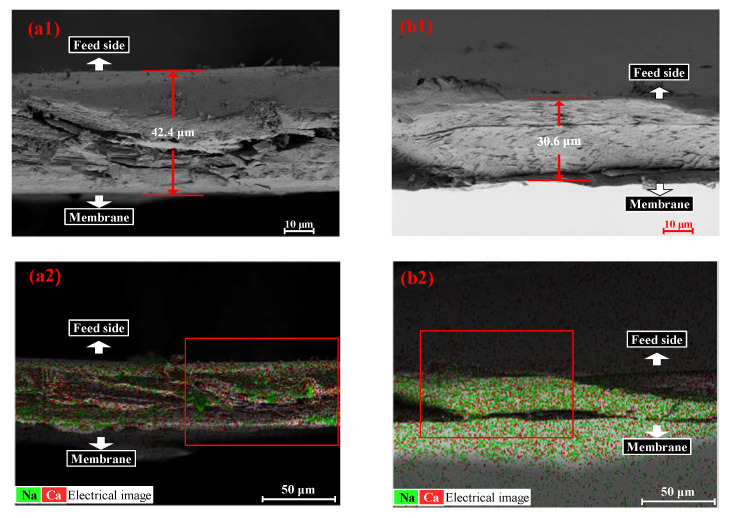
SEM morphology (**a1**,**b1**) and EDS images (**a2**,**b2**) of Ca and Na elements in the cross-sectional profile of CARP. (**a1**,**a2**) are CARP images for 1.0 g/L SA solution as the FS and 1 M CaCl_2_ solution as the DS, where (**a1**) represents the red box in (**a2**); (**b1**,**b2**) are CARP images for 1.0 g/L SA solution as the FS and 3 M CaCl_2_ solution as the DS, where (b1) represents the red box in (**b2**).

**Figure 6 membranes-13-00207-f006:**
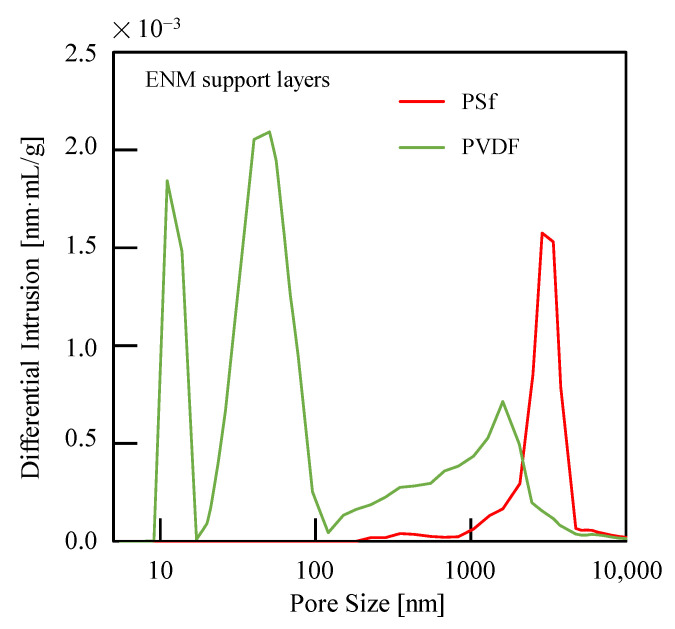
Pore sizes of electrospinning nanofiber membrane (ENM) support layers.

**Figure 7 membranes-13-00207-f007:**
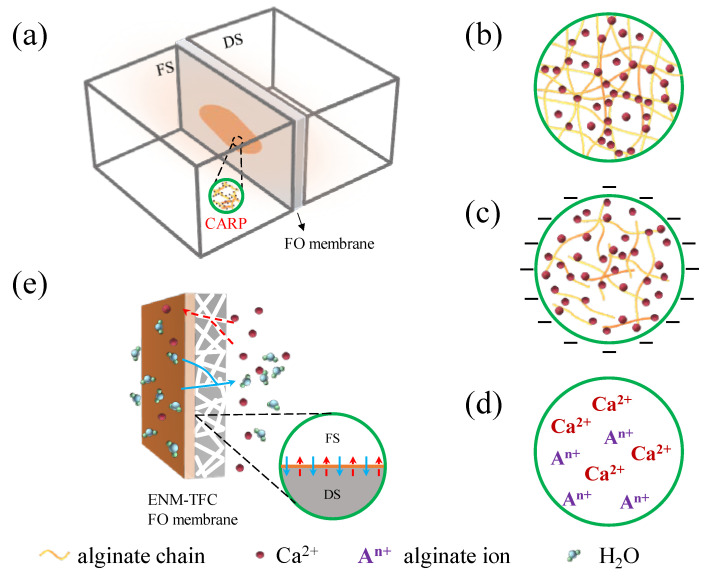
Schematic diagram of the mechanism analysis showing that Ca-Alg recovery production (CARP) affects the reverse diffusion of Ca^2+^ in the draw solution (DS) in calcium alginate production via forward osmosis (FO) with alginate solution as the feed solution (FS). (**a**) CARP on the FO membrane in FO cell; (**b**) image of CARP with molecular sieve role; (**c**) electrification of CARP colloids; (**d**) ions in CARP contributing to osmotic pressure; (**e**) visualization of a thin film composite FO membrane with an electrospinning nanofiber membrane support layer (ENM-TFC FO membrane) regulating water and salt flux.

**Table 1 membranes-13-00207-t001:** Characteristic values in forward osmosis (FO) recovery of Ca-Alg at 5 h of FO. Herein, commercial CTA-ES FO membrane, 1–3 M CaCl_2_ draw solution (DS), 0–3 g/L sodium alginate (SA) feed solution (FS), and 2.5 cm/s flow rates along opposite directions on both sides of the membrane were used.

Conc. of DS	Conc. of SA	*W* _Ca_ ^a^	*W* _Ca,r_ ^b^	*W*_Ca,r_/*W*_Ca_^c^	*W*_Ca_/*W*_SA_^d^	*J* _Ca,av_ ^e^	*J* _w,av_ ^f^
[M]	[g/L]	[mg]	[mg]	[%]	[%]	[g/m^2^/h]	[L/m^2^/h]
1	0	34.85	- ^g^	-	-	1.66 ± 0.12	10.66 ± 0.28
	1	25.55	4.19	16.40	5.11	1.22 ± 0.14	6.63 ± 0.23
2	0	41.43	-	-		1.97 ± 0.11	15.52 ± 0.16
	1	16.99	3.13	18.42	3.40	0.81 ± 0.09	9.30 ± 0.09
	2	35.43	7.92	22.35	3.54	1.69 ± 0.07	7.45 ± 0.13
	3	44.30	16.07	36.28	2.95	2.11 ± 0.05	7.56 ± 0.15
3	0	52.71	-	-	-	2.51 ± 0.21	19.29 ± 0.32
	1	11.59	2.97	25.63	2.32	0.55 ± 0.02	9.58 ± 0.18

^a^ Total content of reverse osmosis Ca^2+^ containing that in the Ca-Alg recovery production (CARP) formed on the FO membrane; ^b^ content of Ca^2+^ in the CARP; ^c^ percentage content of Ca^2+^ in the CARP to total osmosis content of Ca^2+^; ^d^ content rate of total Ca^2+^ reverse osmosis to SA in the FS; ^e^ average flux of total Ca^2+^ reverse osmosis; ^f^ average water flux; ^g^ none.

**Table 2 membranes-13-00207-t002:** Average water flux, *J*_w,av_, and reverse solute flux (RSF), *J*_ca,av_, of Ca^2+^ in forward osmosis (FO) recovery of Ca-Alg after 4 h of FO with various FO membranes. Here, a 1–3 M CaCl_2_ draw solution (DS), 1–3 g/L sodium alginate (SA) feed solution (FS), and the dead-end model (*u* = 3.0 cm/s for the DS side and *u* = 0 cm/s for the FS side) were used.

FO Membranes	Conc. of DS	Conc. of SA	*J* _w,av_	*J* _ca,av_
	[M]	[g/L]	[L/(m^2^·h)]	[g/(m^2^·h)]
CTA-ES	2	1	4.22 ± 0.17	0.39 ± 0.06
PVDF ENM-TFC	2	1	3.98 ± 0.03	0.91 ± 0.09
PSf ENM-TFC	2	1	5.07 ± 0.14	1.13 ± 0.05
	1	1	4.19 ± 0.04	0.91 ± 0.32
	3	1	5.18 ± 0.38	0.63 ± 0.08
	2	2	5.94 ± 0.02	1.30 ± 0.22
	2	3	4.82 ± 0.11	1.11 ± 0.18

**Table 3 membranes-13-00207-t003:** Typical parameters of the ENM support layers fabricated in this study.

ENM Support Layer	Average Pore Size [μm]	Water Contact Angle [°]	Tensile Strength [MPa]	Elastic Modulus [MPa]
PSf	3.90 ± 0.47	42.90 ± 0.54	6.79 ± 0.59	160.51 ± 3.08
PVDF	0.51 ± 0.04	61.60 ± 2.08	6.58 ± 0.80	79.56 ± 2.31

## Data Availability

Not applicable.

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
