# Peer review of "Calcium Alginate Production through Forward Osmosis with Reverse Solute Diffusion and Mechanism Analysis"

_membranes, 2023, doi:10.3390/membranes13020207_

Round 1

Reviewer 1 Report

Paper is written with a good structure. However, It has some errors. I suggest acceptation of the paper after the following revisions:

-Please, eliminate the bulk citations. Example: Your first sentence starts with 7 citations, 6 of them belongs to authors' own papers.

-Figure 2.b You may add R2 values on the graph.

-Figure 3.a. Figure margins are not clear, please state what red line with yellow points stand for on the graph. Also, you should not connect the points with straight line for 3 data points.

-Table 1-2: Standard deviations should be added for flux values.

-Table 3: Standard deviations should be added.

Reviewer 2 Report

This article talks about production of Calcium Alginate via forward osmosis with reverse solute diffusion. This is a novel work and could be of interest to readers in the field of forward osmosis. I recommend certain modifications to the manuscript which has been included in a PDF file attached here.

Round 2

Reviewer 1 Report

Dear Authors,

Thank you for addresing my comments. I suggest ACCEPTION. Just as suggestion, please, avoid cutting membranes for cross section images. Instead, you may freeze with liquid nitrogen and break them for precise cross-section images. 

Best wishes